# Prevalence and determinants of anemia among young (15–24 years) women in Ethiopia: A multilevel analysis of the 2016 Ethiopian demographic and health survey data

**Misganaw Gebrie Worku**[1]*, **Getayeneh Antehunegn Tesema**[2], **Achamyeleh Birhanu Teshale**[2]

1 Department of Human Anatomy, College of Medicine and Health Science, School of Medicine, University of Gondar, Gondar, Ethiopia, 2 Department of Epidemiology and Biostatistics, Institute of Public Health, College of Medicine and Health Sciences, University of Gondar, Gondar, Ethiopia

* misgeb2008@gmail.com

## Abstract

### Background

Anemia is a major global public health problem that had tremendous impacts on human health, social and economic development. African countries contribute to the highest-burden of anemia among women, particularly in adolescent females and young women. Anemia among young women remains a public health problem in most parts of Africa, including Ethiopia. Therefore, this study aimed to investigate the prevalence and determinants of anemia among young women in Ethiopia.

### Methods

A secondary data analysis was conducted based on the 2016 Ethiopian Demographic and Health Survey (EDHS) data. A total weighted sample of 5796 young women (15–24 years) was included in this study. We employed a multilevel analysis to assess factors associated with anemia since the EDHS has hierarchical nature. Deviance, Intraclass Correlation Coefficient (ICC) and Median Odds Ratio (MOR) were used for model comparison as well as for assessing model fitness. Variables with a p-value of less than 0.20 at bivariable multilevel analysis were considered for the multivariable multilevel analysis. In the multivariable multilevel analysis variables with p-value $\leq$ 0.05 were declared to be a significant factor associated with anemia, and the Adjusted Odds Ratio (AOR) with the 95% Confidence Interval (CI) were reported to assess the strength and direction of the association.

### Results

The overall prevalence of anemia among young women was 21.7% (95%CI = 20.7%, 22.8%). In the multivariable multilevel binary logistic regression analysis; being Muslim

**Data Availability Statement:** Data is available online and you can request it from www. measuredhs.com. The authors did not have any

special access privileges that others would not have.

**Funding:** The authors received no specific funding for this work.

**Competing interests:** The authors declare no competing interest.

**Abbreviations:** CI, Confidence Interval; CSA, Central Statistical Agency; DHS, Demographic Health Survey; EA, Enumeration Ar; EDHS, Ethiopian Demographic Health Survey; ICC, Intraclass Correlation Coefficient; LLR, Likelihood Ratio; PCV, Proportional change in Variance; WHO, World Health Organization.

religion follower [adjusted odds ratio (AOR) = 1.31, 95%CI = 1.07, 1.70] and being protestant religion follower [AOR = 1.31; 95%CI = 1.01, 1.71], being rural dweller [AOR = 1.34; 95%CI = 1.02, 1.78], and being married [AOR = 1.46; 95%CI = 1.22, 1.74] were significantly associated with higher odds of anemia among young women. While, modern contraceptive use (AOR = 0.66; 95%CI = 0.53, 0.83) were significantly associated with lower odds of anemia among young women.

## Conclusion

In this study, the prevalence of anemia among young women was high. Being a follower of Muslim and protestant religions, being married women, modern contraceptive use and being from the rural area were found to be significant determinants of anemia among young women. Therefore, giving special attention to these high-risk groups and distributing modern contraceptives for those in need of it could decrease this devastating public health problem in young women.

## Background

Anemia is a global public health problem that affects both developing and developed countries with a high impact on human health, social and economic growth [1]. It is prevalent at all stages of life, but it is more prevalent in female adolescents and young women [1]. Young women including adolescent females are susceptible to anemia because of their biological demands for micronutrients associated with rapid body growth and the depletion of these nutrients due to parasitic infestations [2]. Anemia is also common in puberty due to the onset of menstruation, which alters an individual's iron status by generating more demand for iron, blood loss and pro-inflammatory processes due to menstrual cycles [3].

Globally, it was estimated that one-third of the total population (32.9%) is suffered from anemia with sub-Sahara African countries contributed to the highest anemia burden [4]. More than half of young women worldwide have suffered from anemia and this figure is significantly higher than the World Health Organization's cut-off value for identifying anemia as a public health problem [5]. Approximately one-quarter of young women in developing countries are anemic [6]. Most African and other low and middle-income countries contribute to the highest-burden of anemia among young women [5]. Anemia burden among young women is also common in sub-Saharan African countries which ranges from 13.7% in Ethiopia to 61.5% in Ghana [5]. Ethiopia also shares the high burden of anemia in young women which ranges from 24% to 38%, with an average rate of 29%[7]. Anemia in young women is a serious condition which impedes them from reaching their full potential by reducing educational achievement, labor productivity as well as their cognitive capacity and affect their mental health [1, 8]. Besides, in pregnant women, the risk of birth complications and the delivery of low birthweight infants increases with anemia [9].

According to different studies, many factors such as educational status, marital status, wealth status, nutritional status, occupation, type of toilet facility, source of drinking water, contraceptive use, distance from the health facility, and region are associated with anemia in young women [3, 10, 11]. The extra needs of nutrients because of rapid growth and physical change in young women commonly result in nutritional deficiencies which are the common causes for anemia.

Despite its common occurrence in young women, most previous studies focused on anemia among the reproductive age group (15 to 49 years) [12, 13] and to our knowledge, there is a scarcity of information on the prevalence of anemia among young women and its determinants in Africa including Ethiopia. Anemia due to nutritional deficiency rises at the beginning of puberty and associated physical and physiological changes that occur in adolescents and young women that place a major demand on their nutritional requirements, making them more vulnerable to nutritional deficiency anemia [2]. As many literatures reported, the potential factors affecting reproductive age and young women's anemia are not similar, as young age is the time where nutritional demand is highest. Besides, this study was conducted based on nationally representative Ethiopian Demographic and Health Survey (EDHS) data with a larger sample size that could provide valid information for countries, particularly sub-Saharan African countries and other low- and middle-income countries that had similar socio-economic and socio-cultural patterns. Therefore, this study aimed to investigate the prevalence of anemia and its determinants among young women in Ethiopia. The findings of this study could help to inform policymakers as well as governmental and non-governmental organizations about the magnitude of this problem as well as the potential factors associated with anemia to plan intervention strategies.

## Methods

### Study area

The study was conducted in Ethiopia which is located at the horn of Africa between $3^0$ and $15^0$ north latitude and $33^0$ and $48^0$ east longitude. The country encompasses 1.1 million sq. Km. Its topographic feature ranges from 4550 meters above sea level to 110 meters below sea level. The current population of Ethiopia is estimated to be 115,286,168 and about 30% of this population is in the young age groups. Ethiopia is administratively divided into nine regional states (Tigray, Afar, Amhara, Oromia, Somali, Benishangul-Gumuz, Southern Nations Nationalities and People Region (SNNPR), Gambella and Harari) and two city administrations (Addis Ababa and Dire Dawa) which is again subdivided into 68 zones, 817 districts and 16,253 kebeles (the country's lowest administrative units) (Fig 1).

### Data source

We used the Ethiopian demographic and health survey (EDHS) 2016 data for this study. A multistage stratified cluster sampling technique was employed to select the study participants using the 2007 population and housing census as a sampling frame. In the first stage, a total of 645 Enumeration Areas (EAs) (202 were from urban areas and the rest from rural areas) were selected while in the second stage, a fixed number of 28 households were selected per each EAs. In the EDHS 2016, a total of 16650 households, 12688 men, and 15683 women were successfully interviewed. For this study, a total weighted sample of 5796 young women aged 15 to 24 years was included (Fig 2). The detailed sampling procedure is presented in the EDHS 2016 report [14].

### Study variables

**Outcome variable.** This study was based on hemoglobin level adjusted by altitude and smoking, which was already provided in the EDHS data. The hemoglobin level found in the survey data set was already adjusted for altitude using the adjustment formula (adjust = − 0.032*alt + 0.022*alt2 and adjHb = Hb—adjust (for adjust > 0)). The outcome variable for this study was anemia level among young women (women in the age group of 15–24 years),

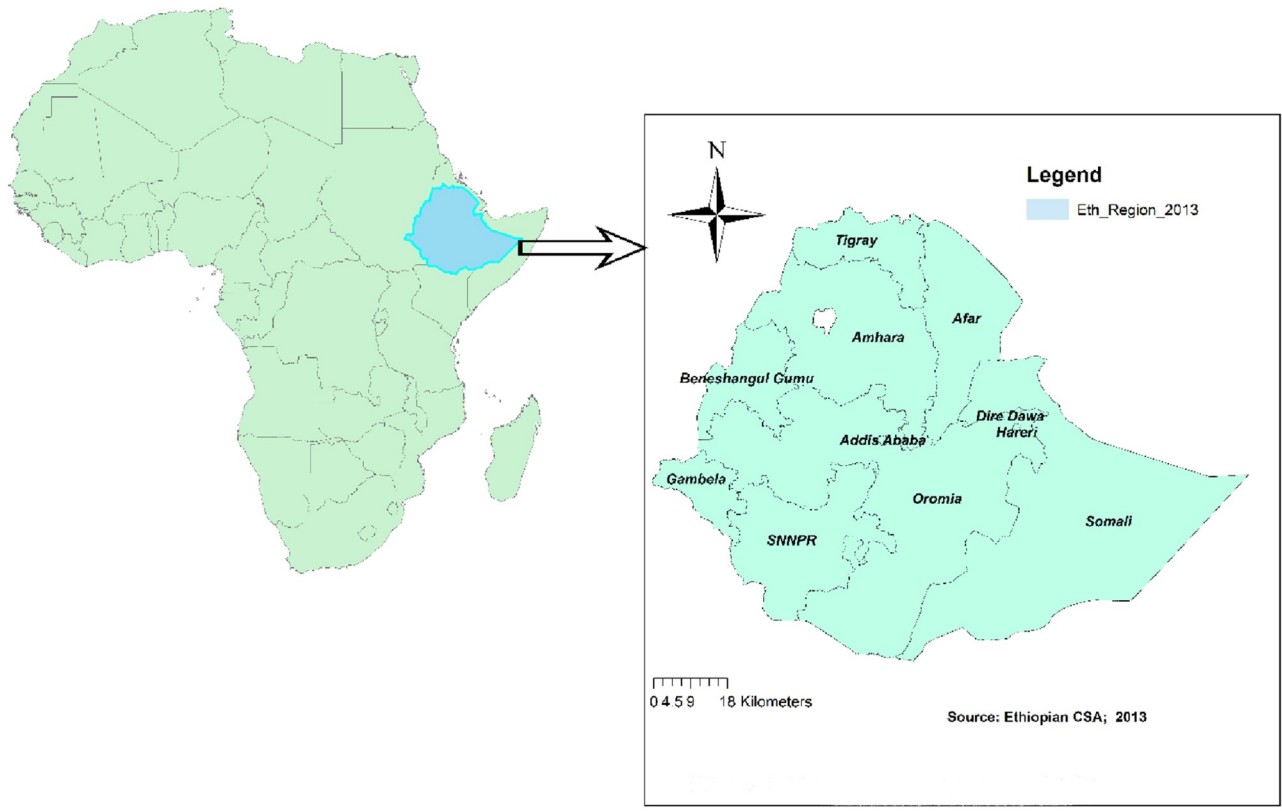

**Fig 1. Map of the study area (using ArcGIS version 10.6 software).**

which was measured based on women pregnancy status; for pregnant women a hemoglobin value of <11 g/dL was considered as anemic and a non-pregnant woman with a hemoglobin value of <12 g/dL was considered anemic [2].

**Independent variables.** For this study, both individual and community-level factors were included as independent variables. The individual-level variables considered for our study were; the age of respondent, educational level, religion, marital status, occupation, wealth status, sex of household head, type of toilet facility, source of drinking water, Body Mass Index (BMI), distance from the health facility, family size, modern contraceptive use, and media exposure. Whereas residence, region and community poverty level were the community level variables included in this study (Table 1).

## Data management and analysis

Data extraction, recoding and analysis (both descriptive and analytical) were done using STATA version 14 software. The data were weighted before any statistical analysis to restore the representativeness of the data and to get a reliable estimate and standard error. A multilevel binary logistic regression analysis was done to identify significant determinants of anemia to consider the hierarchical nature of EDHS data. In EDHS, women were nested within-cluster and women within the same cluster are more likely to share similar characteristics than women in another cluster, which violates the independent assumptions of the standard logistic regression model such as the independent and equal variance assumptions. Four models were constructed; the first model (the null model) constructed only with the presence of outcome variable without independent variables, the second model (Model I) fitted the individual-level

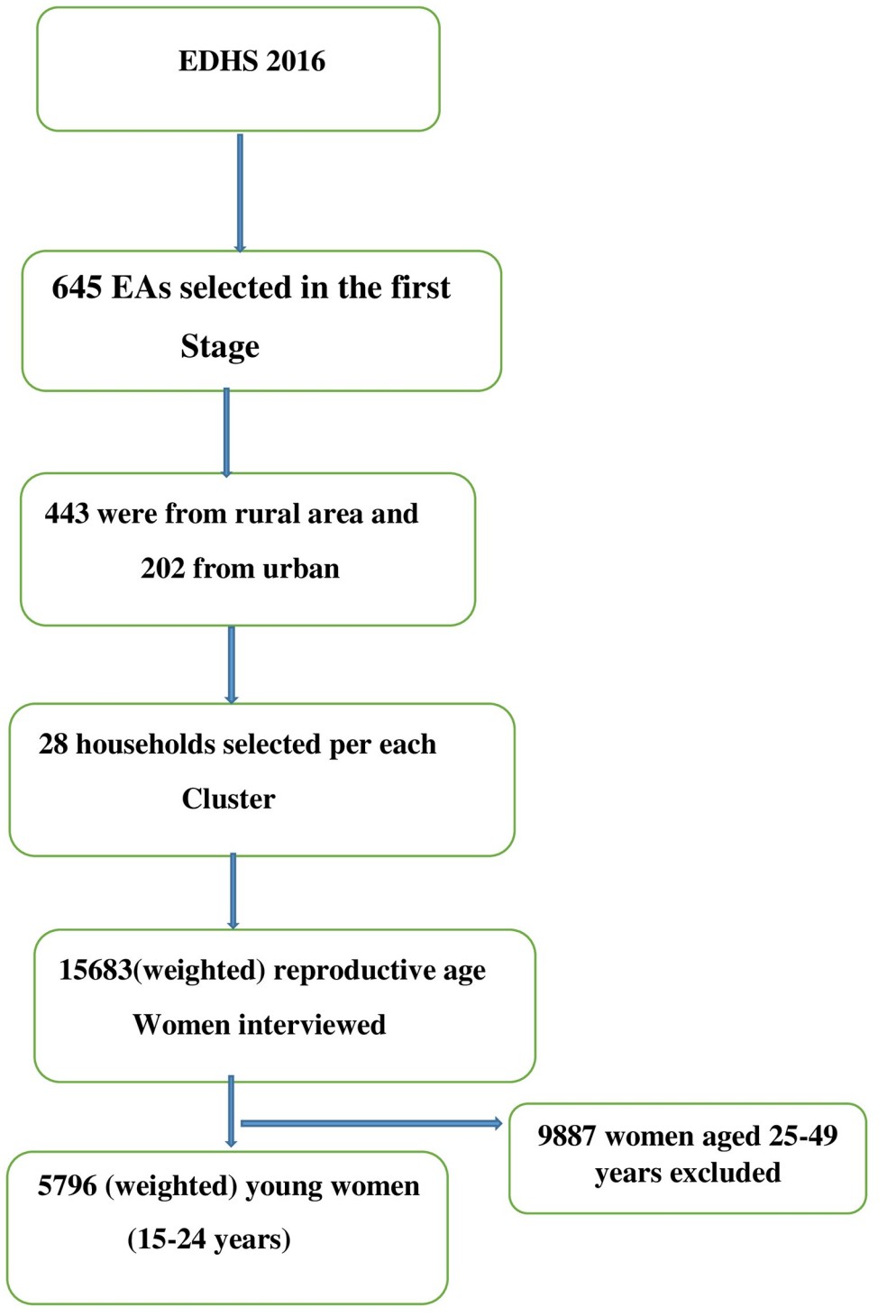

**Fig 2. Flow diagram showing the sampling procedure.**

variables only with the outcome variable, third model (Model II) fitted community-level variables only with the outcome variable, and the final model (model III) fitted both individual and community level variables with the outcome variable. The Intraclass Correlation Coefficient (ICC), and Median Odds Ratio (MOR) were checked to indicate whether there was

**Table 1. Description and measurement of the independent variable.**

| Independent variables and their description/categorization | |
|---|---|
| Individual level variables | |
| Age Group | Current age of the women and re-coded in to two categories with values of "0" for 15–19, "1" for 20–24. |
| Religion | Re-coded in four categories with a value of "0" for Muslim, "1" for Orthodox, "2" for protestant, and "3" for other religious groups (combining catholic, traditional and the other religious categories as most young women in this category are small in number). |
| Wealth Index | The datasets contained wealth index that was created using principal components analysis coded as "poorest", "poorer", "Middle", "Richer", and "Richest in the EDHS data set." For this study we recoded it in to three categories as "poor" (includes the poorest and the poorer categories), "middle", and "rich" (includes the richer and the richest categories) |
| Occupation | Re-coded in two categories with a value of "0" for not working, and "1" for working. |
| Distance to health facility | The variable distance to health facility recorded as a big problem and not a big problem in the dataset was retained without change, which is respondents' perception during the survey whether they perceived the distance from the health facility to get self-medical help as a big problem or not. |
| Media exposure | A composite variable obtained by combining whether a respondent reads newspaper/ magazine, listen to radio, and watch television with a value of "0" if women were not exposed to at least one of the three media, and "1" if a woman has access/exposure to at least one of the three media. |
| Educational status | This is the minimum educational level a woman achieved and re-coded into three groups with a value of "0" for no education, "1" for primary education, and "2" for secondary and above (combining secondary and higher education categories together). |
| Marital status | This was the current marital status of women and recoded in two categories with a value of "0" for unmarried (includes those who were never in union, divorced, widowed, and separated), and "1" for "married" (includes those living with a partner and those who are married) |
| Type of toilet facility | Recoded into two categories as "unimproved "includes and "improved", using the DHS guide. |
| Source of drinking water | By using the DHS guide it was recoded into two categories as "unimproved" and "improved source" |
| Sex of household | The variable sex of household head was recorded as male and female in the dataset and we used without change. |
| Modern contraceptive | Recoded into two categories with a value of 0 for "no" if a woman doesn't use any of the modern contraceptive methods, and 1 for "Yes" if a women use any of the modern contraceptive methods. of either of or combination of the following methods (female sterilization, male sterilization, contraceptive pill, intrauterine contraceptive device, injectables, implants, female condom, male condom, diaphragm, contraceptive foam and contraceptive jelly, lactational amenorrhea method, standard days method, and respondent-mentioned other modern contraceptive methods (including cervical cap, contraceptive sponge,)) |
| Family size | Recoded in to two categories as 1–5, and greater than 5. |
| Body mass index | Re-coded in to three categories with values of 0 for underweight ($<18.5 \text{Kg/m}^2$), 1 for normal (18.5 to 24.9 $\text{Kg/m}^2$), and 2 for overweight ($>25 \text{Kg/m}^2$). |
| Community-level variables | |
| Community poverty level | Measured by proportion of households in the poor (combination of poorer and poorest) wealth quintile derived from data on wealth index. Then it was categorized based on national median value as: low (communities in which $<50\%$ of women had poor socioeconomic status) and high (communities in which $\geq 50\%$ of women had poor socioeconomic status) poverty level. |
| Type of place of residence | The variable place of residence recorded as rural and urban in the dataset was used without change. |
| Region | The variable region was corded in to 11 categories in the dataset and we was retained without change. |

clustering or not. Model comparison/fitness was done using deviance (-2 log-likelihood) and the Proportional Change in Variance (PCV) since these models were nested, and the model with the lowest deviance was chosen. Both bivariable and multivariable multilevel logistic regression were done and variables with p-value <0.2 in the bivariable analysis were considered for multivariable analysis. Finally, variables with P-value <0.05 in the multivariable analysis were considered as significant factors associated with anemia.

### Ethical consideration

Since the study was a secondary data analysis, based on publically available survey data, ethical approval and participant consent were not necessary. However, we asked the DHS Program and permission was granted to download and use the data for this study from http://www.dhsprogram.com. The Institution Review Board approved procedures for DHS public-use datasets do not in any way allow respondents, households, or sample communities to be identified. There are no names of individuals or household addresses in the data files. The document was submitted to the University of Gondar ethical review board (one of the major University in Ethiopia) and the ethical review board approved that ethical clearance is not needed for such type of study, since it is based on nationally representative EDHS data.

## Results

### Sociodemographic characteristics of respondents

**Individual-level factors.** A total sample of 5796 young women was included in this study. More than half (54.61%) of the respondents were aged 15 to 19 years and 54.67% of respondents had primary education. Regarding wealth status and religion, 48.09% of respondents were from rich households and about 43.01% of respondents were practicing orthodox Christian religion. About 64.18% of respondents were from households with an improved water source and 83.78% of participants were from households with an unimproved toilet facility. Concerning family size, the majority (58.99%) of women were from a family size of 1 to 5 and 72.59% of women were not currently working (Table 2).

**Community-level factors.** The majority (77.03%) of the respondent were rural dwellers and 36.46% of respondents were from the Oromia region. More than half (51.14%) of the participants were from communities with a higher poverty level (Table 2).

### Prevalence of anemia among young women in Ethiopia

In this study, the prevalence of anemia among young women was 21.7% (95% CI: 20.7%, 22.8%). Young women from Somalia (56.80%) and Afar (43.93%) region had a higher prevalence of anemia and those from Addis Ababa had a lower prevalence of anemia (Fig 3).

### Random effect model and model fitness

The random-effect model was assessed using ICC, MOR, and PCV. In the null model the ICC value which was 0.22, indicates that 22% of the total variation of anemia in young women was due to differences between clusters/communities. Besides, the highest MOR value which was 2.53 indicates that there was significant clustering of anemia in young women. Moreover, the highest PCV (0.72) in the final model (model III) revealed that about 72% of the variation in anemia was explained by both individual and community-level factors. Regarding model fitness, the final model (model III), which incorporates both individual and community level factors, was the best-fitted model since it had the lowest deviance (5945.24) (Table 3).

**Table 2. Sociodemographic characteristics of the respondents in Ethiopia, 2016 (N = 5796).**

| Variables | | Frequency (%) |
|---|---|---|
| **Individual-level factors** | | |
| Age (years) | 15–19 | 3165(54.61%) |
| | 20–24 | 2631(45.39%) |
| Highest education level | No education | 1164(20.08%) |
| | Primary education | 3168(54.67%) |
| | Secondary and above | 1464(25.25%) |
| Wealth index | Poor | 1932(33.33%) |
| | Middle | 1077(18.58%) |
| | Rich | 2787(48.09%) |
| Occupation | No | 4207(72.59%) |
| | Yes | 1589(27.41%) |
| Religion | Muslim | 1758(30.33%) |
| | Orthodox | 2493(43.01%) |
| | Protestant | 1416(24.43%) |
| | Other * | 129(2.23%) |
| Marital statues | Unmaried | 3591(61.97%) |
| | Maried | 2205(38.03%) |
| Family size | 1–5 | 3361(58.99%) |
| | >5 | 2435(42.01%) |
| Body mass index | Underweight | 1405(24.24%) |
| | Normal | 4129(71.24%) |
| | Overweight | 262(4.52%) |
| Type of toilet facility | Unimproved | 4856(83.78%) |
| | Improved | 940(16.22%) |
| Source of drinking water | Unimproved | 2076(35.82%) |
| | Improved | 3720(64.18%) |
| Modern contraceptive use | No | 943(16.27%) |
| | Yes | 4853(83.73%) |
| Sex of household head | Male | 4388(75.71%) |
| | Female | 1408(24.29%) |
| Media exposure | No | 2897(49.98%) |
| | Yes | 2899(50.02%) |
| Distance to health facility | Big problem | 2814(48.54%) |
| | Not big problem | 2982(51.46%) |
| **Community level factors** | | |
| Residence | Urban | 1331(22.97%) |
| | Rural | 4465(77.03%) |
| Community poverty level | Low | 2832(48.86%) |
| | High | 2964(51.14%) |
| Region | Tigray | 465(8.03%) |
| | Afar | 52(0.90%) |
| | Amhara | 1340(23.12%) |
| | Oromia | 2113(36.46%) |
| | Somali | 162(2.81%) |
| | Beni Shangul | 58(1.01%) |
| | SNNPR | 1180(20.36%) |
| | Gambella | 17(0.30%) |
| | Harari | 13(0.23%) |
| | Addis Ababa | 362(6.25%) |
| | Dire Dawa | 32(0.54%) |

* = Catholic, traditional, other.

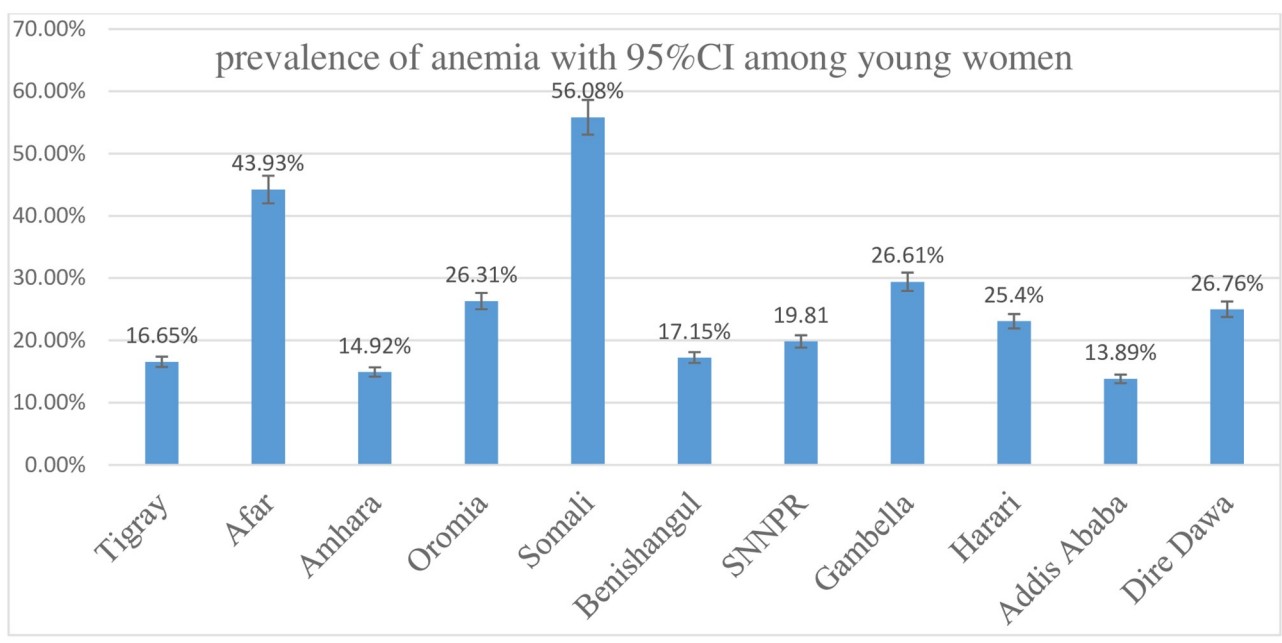

**Fig 3. Anemia prevalence by region among young women in Ethiopia; 2016.**

## Determinant of anemia among young women in Ethiopia

We used the final model (the best-fitted model) to assess the determinants of anemia among young women in Ethiopia.

All variables except sex of the household head were significant in the bivariable analysis (had p<0.20). In the multivariable analysis, both individual-level factors (religion, marital status and modern contraceptive use) and community-level factors (residence and region) were found to be significant determinants of anemia among young women. The odds of having anemia were 1.31 [adjusted odds ratio (AOR) = 1.31; 95%CI = 1.07, 1.70], and 1.31 [AOR = 1.3; 95%CI 1.01, 1.71] times higher among women who practiced Muslim and protestant religions respectively as compared with those who practicing orthodox Christian religion. The odds of developing anemia in young women was 1.46 [AOR = 1.46; 95%CI = 1.22, 1.74] times higher among married women as compared with their counterparts. A young woman who used modern contraceptives had 34% (AOR = 0.66; 95%CI = 0.53, 0.83) lower odds of developing anemia compared with those who do not use modern contraceptives. Being a woman from a rural area had 1.34 [AOR = 1.34; 95%CI = 1.02, 1.78] times higher odds of anemia as compared with those from urban. The odds of developing anemia were higher among women in

**Table 3. Random effect model and model fitness for the assessment of anemia among young women in Ethiopia.**

| Parameter | Null model | Model I | Model II | Model III |
|---|---|---|---|---|
| Community-level variance | 0.946 | 0.410 | 0.305 | 0.015 |
| ICC | 0.22 | 0.11 | 0.08 | 0.07 |
| MOR | 2.53(2.27–2.80) | 1.84(1.66–2.03) | 1.70(1.54–1.88) | 1.63(1.48–1.82) |
| PCV | Reff | 0.57 | 0.68 | 0.72 |
| **Model fitness** | | | | |
| Log likelihood | -3156.66 | -3029.84 | -3005.15 | -2972.62 |
| Deviance | 6313.32 | 6059.68 | 6010.30 | 5945.24 |

Somali [AOR = 3.63; 95% CI = 2.39, 5.51], Dire Dawa [AOR = 1.52; 95% CI = 1.01, 2.30], and Afar (AOR = 2.02; 95%CI = 1.31, 3.11) regions and lower among women from Amhara (AOR = 0.60; 95%CI = 0.39, 0.92) and Beni-shangul (AOR = 0.60; 95%CI = 0.37, 0.93) regions as compared with women from Addis Ababa (Table 4).

## Discussion

Anemia among young women is a major public health problem in low and middle-income countries [5]. This study aimed to investigate the prevalence and determinants of anemia among young women in Ethiopia. In this study, the prevalence of anemia among young women was 21.70% (95% CI: 20.66%, 22.78%), which is in agreement with other studies conducted in Armenia and East Timor [5]. The prevalence in this study was greater than a previous study in Ethiopia that found anemia in 13.7% of young women [5]. This might indicate the poor management and implementation of health policy in Ethiopia and also the variation of anemia prevalence across population subgroups. Also, this could be due to the increased risk of chronic disease and other pathological conditions that may increase the risk of anemia over time [15]. The finding in this study was also greater than the anemia prevalence in Rwanda, with greater than the 15.6% anemia found among young women in Rwanda [5]. This might be because of the socioeconomic and sociocultural differences between populations of different African countries. However, the prevalence in our study was smaller than studies reported in sub-Saharan African countries (Burkina Faso, Benin and democratic republic of Congo) [5]. This might be due to the variation in the availability of foodstuffs, health care services access and utilization, and living conditions [16].

The multilevel logistic regression analysis showed that young women who practice Muslim and protestant religion, married, rural residence, modern contraceptive use, and being women from Afar, Amhara, Somalia, Beni-shangul, and Dire Dawa were significantly associated with anemia. The odds of developing anemia among women who practice Muslim religion were higher as compared with Orthodox Christian religion followers. This is consistent with a previous study done in Ethiopia [11]. The possible explanation could be due to the deeply rooted cultural and religious beliefs in food restrictions like pork meat, fish meat, goat meat and bacon and other potential dietary nutrients that are the best source of iron, vitamin B12 and folate which are not allowed to be used culturally in Muslim religions [17]. Besides, the increased chances of anemia among Muslim religious followers might be due to the restriction of the use of different hormonal contraceptives, which could minimize the risk of developing anemia [18]. Married young women had higher odds of anemia as compared to unmarried women. This is in contrast with the finding of prior studies [3, 13, 19]. This may be because married women can give birth and are vulnerable to pregnancy and birth-related bleeding, as well as complications that may raise the risk of anemia [20].

The study at hand also revealed that being women from rural areas was associated with a higher likelihood of anemia. This is in agreement with a prior study done in Ethiopia [3]. This might be attributed to inadequate access to dietary supplements, due to limited access to maternal health care services, in rural women [12]. In addition, parasitic infections such as hookworm and malaria are widespread in rural residents due to their lifestyles (such as walking barefooted and poor personal hygiene) which are the leading causes of anemia in Ethiopia [21]. Women who used modern contraceptives had lower odds of developing anemia as compared to those who didn't use modern contraceptives. This finding was supported by a study conducted in Rwanda [22]. This could be justified by the protective effects of modern contraceptives on menstrual bleeding, pregnancy and birth-related hemorrhages [23]. Concurrent

**Table 4. Bi variable and multivariable multilevel analysis for the assessment of determinants of anemia among young women in Ethiopia, 2016.**

| Variables | | Anemia | | COR (95%CI) | AOR (95%CI) | p- value |
|---|---|---|---|---|---|---|
| | | Yes | No | | | |
| Age (years) | 15–19 | 631 | 2534 | 1 | 1 | |
| | 20–24 | 627 | 2004 | 0.06(1.01, 1.31) | 1.07(0.90, 1.26) | 0.34 |
| Highest education level | No education | 325 | 839 | 1 | 1 | |
| | Primary education | 699 | 2469 | 0.57(0.48, 0.68) | 0.93(0.77, 1.12) | 0.45 |
| | Secondary and above | 234 | 1230 | 0.40(0.32, 0.50) | 0.81(0.65, 1.03) | 0.09 |
| Wealth index | Poor | 524 | 1408 | 1 | 1 | |
| | Middle | 220 | 857 | 0.62(0.50, 0.78) | 0.89(0.70, 1.12) | 0.34 |
| | Rich | 514 | 2273 | 0.46(0.38, 0.54) | 0.85(0.66, 1.08) | 0.18 |
| Occupation | No | 956 | 3251 | 1 | 1 | |
| | Yes | 302 | 1287 | 0.74(0.63, 0.86) | 0.93(0.79, 1.09) | 0.40 |
| Religion | Orthodox Christian | 399 | 2094 | 1 | 1 | |
| | Muslim | 497 | 1261 | 3.07(2.56, 3.70) | 1.31(1.07, 1.70) | 0.01 |
| | Protestant | 307 | 1109 | 1.53(1.21, 1.92) | 1.31(1.01, 1.71) | 0.83 |
| | Other * | 55 | 74 | 1.71(0.93, 3.15) | 1.30(0.71, 2.36) | 0.90 |
| Marital status | Unmaried | 676 | 2915 | 1 | 1 | |
| | Maried | 582 | 1623 | 1.58(1.37, 1.81) | 1.46(1.22, 1.74) | 0.00 |
| Family size | 1–5 | 740 | 2621 | 1 | 1 | |
| | >5 | 519 | 1916 | 0.99(0.87, 1.14) | 1.08(0.93, 1.25) | 0.30 |
| Body mass index | Normal | 885 | 3244 | 1 | 1 | 0.92 |
| | Underweight | 320 | 1085 | 1.10(0.95, 1.28) | 1.01(0.86, 1.17) | |
| | Overweight | 53 | 209 | 0.71(0.52, 0.98) | 0.79(0.57, 1.09) | 0.15 |
| Type of toilet facility | Unimproved | 1091 | 3765 | 1 | | |
| | Improved | 167 | 773 | 0.74(0.62, 0.89) | 0.86(0.71, 1.06) | 0.18 |
| Source of drinking water | Unimproved | 539 | 1534 | 1 | 1 | |
| | Improved | 719 | 3001 | 0.69(0.59, 0.81) | 0.98(0.83, 1.16) | 0.87 |
| Modern contraceptive use | No | 1089 | 3764 | 1 | 1 | |
| | Yes | 170 | 773 | 0.66(0.53, 0.82) | 0.66(0.53, 0.83) | 0.00 |
| Media exposure | No | 2179 | 2179 | 1 | 1 | |
| | Yes | 2358 | 717 | 0.58(0.51, 0.68) | 0.88(0.74, 1.04) | 0.16 |
| Distance to health facility | Big problem | 677 | 2136 | 1 | 1 | |
| | Not big problem | 581 | 2401 | 0.76(0.65, 0.88) | 0.97(0.83, 1.13) | 0.74 |
| Residence | Urban | 219 | 1112 | 1 | 1 | |
| | Rural | 1034 | 3425 | 2.01(1.61, 2.48) | 1.34(1.10, 1.78) | 0.03 |
| Community poverty level | Low | 524 | 2308 | 1 | 1 | |
| | High | 735 | 2229 | 2.22(1.83, 2.70) | 0.90(0.70, 1.15) | 0.41 |
| Region | Addis Ababa | 50 | 312 | 1 | 1 | |
| | Tigray | 77 | 388 | 1.25(0.86, 1.83) | 0.81(0.53, 1.22) | 0.31 |
| | Afar | 23 | 29 | 5.80(3.97, 8.49) | 2.02(1.31, 3.11) | 0.001 |
| | Amhara | 200 | 1140 | 1.03(0.69, 1.52) | 0.60(0.39, 0.92) | 0.02 |
| | Oromia | 556 | 1557 | 2.18(1.52, 3.13) | 0.95(0.63, 1.43) | 0.82 |
| | Somali | 91 | 72 | 9.73(6.70, 14.11) | 3.63(2.39, 5.51) | 0.00 |
| | Beni Shangul | 10 | 48 | 1.23(0.80, 1.90) | 0.60(0.37, 0.93) | 0.02 |
| | SNNPR | 234 | 942 | 1.40(0.96, 2.04) | 0.67(0.43, 1.03) | 0.07 |
| | Gambella | 5 | 13 | 2.37(1.58, 3.57) | 1.19(0.78, 1.85) | 0.43 |
| | Harari | 3 | 10 | 2.11(1.33, 3.24) | 1.19(0.76, 1.84) | 0.43 |
| | Dire Dawa | 8 | 23 | 2.37(1.57, 3.57) | 1.52(1.01, 2.30) | 0.04 |

* = Catholic, traditional, other.

iron supplementation is also available particularly for those women who have used oral contraceptives, which is very important for the prevention of anemia [23].

Moreover, in this study, the probability of developing anemia differed across regions, this may be attributed to differences in the source of food and difference in the availability of the varieties of foods in different regions [24]. Additionally, the difference in anemia risk across regions may be related to the availability and use of health services, and the disparity in the distribution of communicable diseases (such as malaria, visceral leishmaniasis, and hookworm, which are common in lowland areas) across different regions, which are considered as the most common causes of anemia. Furthermore, the regional variation of anemia may be related to the low socio-economic status of some regions especially those found in border areas of Ethiopia, which may lead to inadequate access to foods that are rich in iron [11].

## Strength and limitation of the study

The study has many strengths, first, the study was based on a weighted nationally representative data with a large sample size. Also, the analysis was done using the multilevel analysis to accommodate the hierarchical nature of the EDHS data to get a reliable estimate. Moreover, since it is based on the national survey data, the study has the potential to give insight for policy-makers and program planners to design appropriate intervention strategies both at national and regional levels. However, this study had limitations in that the EDHS survey was based on respondents' self-report, this might have the possibility of recall bias. Besides, since this study was based on survey data, we are unable to show the temporal relationship between anemia and independent variables. Moreover, since this study was based on information available on the survey, other confounders such as infections (the presence of malaria, intestinal parasites, and HIV/AIDS) were not adjusted. Moreover, it is difficult to assess some important variables such as age at first childbearing and any prophylaxis given during pregnancy as this information is recorded for those who gave birth and those who were/are pregnant respectively (missing for those who did not give birth and not pregnant).

## Conclusion

In this study, the prevalence of anemia among young women was high. Both individual and community-level factors were associated with anemia in young women. Being a follower of Muslim and protestant religions, being married women, modern contraceptive use, being from rural area and region were found to be significant determinants of anemia among young women. Therefore, giving special attention to these high-risk groups such as rural dwellers and those in border regions, as well as distributing modern contraceptives for those in need of it could decrease this devastating public health problem in young women.

## Acknowledgments

We greatly acknowledge MEASURE DHS for granting access to the Ethiopia Demographic and Health Surveys data.

## Author Contributions

**Conceptualization:** Misganaw Gebrie Worku, Getayeneh Antehunegn Tesema, Achamyeleh Birhanu Teshale.

**Data curation:** Misganaw Gebrie Worku, Getayeneh Antehunegn Tesema, Achamyeleh Birhanu Teshale.

**Formal analysis:** Misganaw Gebrie Worku, Getayeneh Antehunegn Tesema, Achamyeleh Birhanu Teshale.

**Funding acquisition:** Misganaw Gebrie Worku, Getayeneh Antehunegn Tesema, Achamyeleh Birhanu Teshale.

**Investigation:** Misganaw Gebrie Worku, Getayeneh Antehunegn Tesema, Achamyeleh Birhanu Teshale.

**Methodology:** Misganaw Gebrie Worku, Getayeneh Antehunegn Tesema, Achamyeleh Birhanu Teshale.

**Project administration:** Misganaw Gebrie Worku, Getayeneh Antehunegn Tesema, Achamyeleh Birhanu Teshale.

**Resources:** Misganaw Gebrie Worku, Getayeneh Antehunegn Tesema, Achamyeleh Birhanu Teshale.

**Software:** Misganaw Gebrie Worku, Getayeneh Antehunegn Tesema, Achamyeleh Birhanu Teshale.

**Supervision:** Misganaw Gebrie Worku, Getayeneh Antehunegn Tesema, Achamyeleh Birhanu Teshale.

**Validation:** Misganaw Gebrie Worku, Getayeneh Antehunegn Tesema, Achamyeleh Birhanu Teshale.

**Visualization:** Misganaw Gebrie Worku, Getayeneh Antehunegn Tesema, Achamyeleh Birhanu Teshale.

**Writing – original draft:** Misganaw Gebrie Worku, Getayeneh Antehunegn Tesema, Achamyeleh Birhanu Teshale.

**Writing – review & editing:** Misganaw Gebrie Worku, Getayeneh Antehunegn Tesema, Achamyeleh Birhanu Teshale.

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
