## [Decision Letter · Decision Letter 0]

5 Aug 2020

PONE-D-20-18515

Prevalence and determinants of Anemia among young (15-24 years) women in Ethiopia; A multilevel analysis of the 2016 Ethiopian Demographic and Health Survey data

PLOS ONE

Dear Dr. Worku,

Thank you for submitting your manuscript to PLOS ONE. After careful consideration, we feel that it has merit but does not fully meet PLOS ONE’s publication criteria as it currently stands. Therefore, we invite you to submit a revised version of the manuscript that addresses the points raised during the review process.

Five experts in the field handled your manuscript. We are very thankful for their time and efforts. Although some interest was found in your study, several major comments arose that overshadowed this enthusiasm. These concerns include the overall impact on global policy changes, and perhaps the article should be directed toward national policy makers. Furthermore, there are details about the methods that need to be clarified; the data presentation needs work; and there are suggestions to increase the readability of this manuscript with the necessity to employ an expert to correct the English grammar and syntax.

We look forward to receiving your revised manuscript.

Kind regards,

Frank T. Spradley

Academic Editor

PLOS ONE

2. Thank you for submitting the above manuscript to PLOS ONE. During our internal evaluation of the manuscript, we found significant text overlap between your submission and the following previously published works, some of which you are an author.

https://bmcpregnancychildbirth.biomedcentral.com/articles/10.1186/s12884-020-03024-5

https://www.researchsquare.com/article/rs-7120/v1

Please revise the manuscript to rephrase the duplicated text, cite your sources, and provide details as to how the current manuscript advances on previous work. Please note that further consideration is dependent on the submission of a manuscript that addresses these concerns about the overlap in text with published work.

"None"

4. Your ethics statement must appear in the Methods section of your manuscript. If your ethics statement is written in any section besides the Methods, please move it to the Methods section and delete it from any other section. Please also ensure that your ethics statement is included in your manuscript, as the ethics section of your online submission will not be published alongside your manuscript.

Reviewers' comments:

Reviewer's Responses to Questions

**Comments to the Author**

1. Is the manuscript technically sound, and do the data support the conclusions?

Reviewer #1: Yes

Reviewer #2: Partly

Reviewer #3: Yes

Reviewer #4: Yes

Reviewer #5: Yes

2. Has the statistical analysis been performed appropriately and rigorously? 

Reviewer #1: Yes

Reviewer #2: I Don't Know

Reviewer #3: Yes

Reviewer #4: Yes

Reviewer #5: Yes

3. Have the authors made all data underlying the findings in their manuscript fully available?

Reviewer #1: Yes

Reviewer #2: No

Reviewer #3: Yes

Reviewer #4: Yes

Reviewer #5: Yes

4. Is the manuscript presented in an intelligible fashion and written in standard English?

Reviewer #1: Yes

Reviewer #2: No

Reviewer #3: Yes

Reviewer #4: No

Reviewer #5: No

5. Review Comments to the Author

Reviewer #1: This is a fine paper and will be useful to the field, however I have some comments:

1) The manuscript need English editing.

2) In table 3; it will be better if author add extra column for p value.

Reviewer #2: Prevalence and determinants of Anaemia among young (15-24 years) women in Ethiopia; A multilevel analysis of the 2016 Ethiopian Demographic and Health Survey data. This study is an overview demographic data from Ethiopia were association with anaemia and demographics are made.

I have two major concerns:

As these outcome are interesting and important for national policy makers, I am doubting if it is eligible for the international readers of PLOSONE this as outcomes might be region specific.

Secondly I do not understand why this study is needed, with regards to this age group, are young woman not part of the reproductive age group? What makes them different?

Abstract:

Background:

Line 28: Is this for all of Africa or only sub-sharan Africa? As I believe the needs are not as much possible in northern Africa. Please rephrase.

Line 30:I do not understand the difference between young woman and and woman in the reproductive age group. Is this not the same group? Why is there a difference?

Method:

Line 34 what kind of determinants?

Line 35 MOR/ ICC; please no abbreviations in the abstract without explanation.

Line 36 double (..) .

Line 36-38 this can be shortened not all needed in my perspective in the abstract. What is the definition of young woman’s? I can’t find that In the method.

Results/conclusion: strong and usefull conclusion.

Main manuscript:

Background: The terms young woman and adolescents are used all over the manuscript. However it is unclear what the exact definitions are.

Line 75-78 unclear sentences, which cut-off? Please define more specific.

Line79: Which population? Please define more specific.

Overall line 75-87 this alinea needs to be restructured, the data presenting on adolescents and young woman reproductive are not used in a structured way. Please rephrase and structure.

Line 89-92: This is a line-up most likely from a previous study. Please summarise the only needed information.

Line 93: This is a repetition of what is said in the first alinea. Please rephrase and restructure.

Line 95-96: the reason for his study is that most study look to the reproductive woman, it is unclear for me why this young woman are not part of that see previous comments. I addition: UDAID data 2000 shows that 16% of the woman between 15-19 years have been mothers. Moreover Adolescent childbearing trends and sub-national variations in Ethiopia: a pooled analysis of data from six surveys Yared Mekonnen et al. BMC Pregnancy and Childbirth volume 18, Article number: 276 (2018) shows similar data. Can the author explain why this study is needed next to a study on woman in the reproductive age ?

Line 99: Abbreviation not explained.

Methods:

Line 107-111 can this be displayed in a graph, that makes it easier for the reader to understand.

Line 113: add abbreviation (EDHS) as it used in the next sentence.

Line 113-126 can this be added in a graph/ figure with a flow diagram. That makes it easier for the reader to oversee what happened.

Outcome variable line 128: please define young woman.

Line 129-131 I do not understand this sentence and I am questioning if the sentence is needed. Moreover there is no difference between in mild-moderate-and severe anaemia in the study. So the author should remove this sentence.

Line 132: Reference 14; is this correct? And please add in writing which adjustment is done.

Line 133: Reference 13; correct? This is the WHO definition of anaemia please add the WHO as reference. Independent variable: line 136-140: please add definitions of the variable or add a reference. For example what is wealth status? What includes media exposure?? What includes modern contraceptives? Please rephrase.

Line 138 typo double (,,). Data management and analysis: 148-149 Descriptive (no needed information) please take out.

Line 152-154: this is a difficult statistical method, I cannot comment on it, as I am not familiar with the method.

Results:

Overall percentage can be given in one decimal.

Line 164: weighted sample not needed- please take out. The age range 15-24 can be taken out here, as this will be added to the method section.

The structure of line 165-173: could the author structural outcomes better. As a suggestion group in individual medical- social-economics and community. And use this structure as well in table 1.

Line 181-184: is this difference significant? Figure 1: what is the yellow line, please adjust figure.

Line 186-194: I can not comment on this, as my I do not have enough knowledge on statistics to do so.

Line 196-213/ Table 3: regions: why is all compared to Addis? Nutrition status: why all compared to malnutrition and not to none? Why with occupation display no below yes. Same for contraceptive and media exposure. Please display in a constituent manner. Distance to health facility: not a big problem? What does that mean? How is this defined??

Discussion:

I miss out in the discussion very clear statements why this outcome is 1) different than other woman previous investigated in Ethiopia? 2) is that surprising or not? What did others find in SSA for these age groups? Are you in line with that and if yes or no what are the difference? What are the clear limitations: limitations are stated very minimal. Conclusion: the conclusion is strong.

Line 217: Armenia-Timor: why a comparison tot his places?

Line 218: our country, what was the rate?

Line 219: Rate of comorbidity? This comment is based on which knowledge please explain

Line 236-249: this alinea has a lot of repetitive statements. Please rephrase and structure.

Line 247-249: Please add this to the section on rural areas in line 237-238

Line 254 ( see comment line 236-249)

Imitations: There is no data on age of first child, time between sampling and delivery. So major presumptions are made, but ideally this information would be available to say more about why are these women at risk. Or data on nutrition habits to underline the statements and suggestion. Moreover there is no data on malaria prevalence in this group. Or the availability of prophylaxis during pregnancy. All factors, which can be of major influence. The limitations need rephrasing and suggested limitations should be added.

Reviewer #3: This article covers an important global health concern. However, I believe the manuscript will benefit from a comprehensive language editing.

Please pay particular attention to the following and review the sentences:

Lines 25, 75, 95, 113, 129, 132,140, 150, 167. 170, 199, 229, 234, 238, 244, 253 and 257

Reviewer #4: Great work with potential impact on policy change.

However, the discussion needs to be written with clarity and better understanding. Most of the explanations given for the reported findings are difficult to comprehend.

There are also too many grammatical and spelling errors. The manuscript will benefit from review by a native English speaker.

Reviewer #5: Prevalence and determinants of Anemia among young (15-24 years) women in Ethiopia; A multilevel analysis of the 2016 Ethiopian Demographic and Health Survey data

The authors present findings from an analysis of data to determine the prevalence of anemia young women who participated in the 2016 Demographic Health Survey in Ethiopia. The key findings are the prevalence and identification of determinants of anemia among the study participants. The main strength of the study is the large sample size of women whose data were analyzed. However, the paper has several issues that the authors need to address before the paper can be suitable for publication in the journal.

The first issue is that the paper would need substantial English language editing; there are many editorial errors in the paper. The specific issues that require revision in the different components of the paper are provided below:

Abstract

1. The statement that there is a high prevalence of anemia in young persons (line 29) as a justification for the study is contradictory because If there is already high prevalence of anemia among young women why have the authors conducted another study among this same population? This should be clarified.

2. The statement ‘The overall prevalence of anemia among young women were 21.7% (line 40) should read ‘…was 21.7%’.

Background

1. There is need to clarify that the vulnerable population being referred to in line 69 are female adolescents and young women; this is necessary because male adolescents also suffer from anemia. This point should be clarified throughout the manuscript.

2. The statement ‘over half of young women worldwide are suffered...’ line 75 should read ‘… have suffered’

3. The statement ‘…and African as well...’ line 76 is not clear and should be revised.

4. If the statement ‘approximately one quarter of adolescents…’ line 78 is referring to female adolescents this should be clarified

5. There are many repetitive statements about the fact that sub-Sahara Africa has contributed to the high burden of anemia, this should be revised.

6. The authors have listed the socio-demographic factors (lines 89-92) that contribute to anemia in Ethiopia, but examples are not provided to illustrate this point. Some examples should be provided to illustrate this point.

7. The statement ‘Due to rapid growth…’ line 93 is a repetition because this point has been made earlier

8. The sentence ‘…and up to our knowledge’ line 96 should read ‘…to our knowledge’

9. The data for the study were extracted from the 2016 EDHS; is this the latest survey in the country? Have there been other surveys since the one in 2016? If there has been a more recent survey, the authors need to justify why they have used the 2016 survey as source of data for this study. This is should be clarified

Methods

1. An important information missing about the study area is the population of Ethiopia and more importantly the population of young persons in the country.

2. The reference to altitude and smoking (line 132) is not clear; how do these variables relate to the issue investigated. This should be clarified.

Results

1. The statement ‘of women were not had work’ line 170-171 is not clear and should be revised

2. The word ‘afar’ line 183 should read ‘Afar’

3. The use of the phrase ‘followers of Muslim and protestant religion’ line 203 are not clear. A better phrase may be people who practice Islam or Moslems and protestant Christians. This should be revised.

4. The statement ‘… with their counterparts’ line 207 should read ‘with their counterparts who do not’

5. The figure showing prevalence of anemia should have a number and reference should be made to it in the text

Discussion

1. I suggest that the authors provide the figure being referred to in the statement ‘but greater than the previous reported in our country’ line 218

2. The authors should give examples of the food types being referred to in line 228 and support this statement with appropriate reference

3. The statement i.e. barefooted line 241 should be revised to read walking barefooted

6. PLOS authors have the option to publish the peer review history of their article (what does this mean?). If published, this will include your full peer review and any attached files.

Reviewer #1: No

Reviewer #2: No

Reviewer #3: No

Reviewer #4: **Yes: **KEHINDE OKUNADE

Reviewer #5: **Yes: **Ademola J. Ajuwon

---

## [Author Response · Author response to Decision Letter 0]

25 Aug 2020

Date: August 2020

Author's point to point response to editor and reviewers comments

Title: Prevalence and determinants of Anemia among young (15-24 years) women in Ethiopia; A multilevel analysis of the 2016 Ethiopian Demographic and Health Survey data

Manuscript number: PONE-D-20-18515 

Subject: Submitting a revised version of the manuscript

We would like to thank the reviewers and editor for sharing their view and novel scholarly experiences. The comments are very imperative which we strongly believe in improving the manuscript. We try to address all the comment raised by the revivers and academic editor line by line in the main document. The point-by-point responses for each of the comments, questions, and the revised manuscript are provided in the attached documents.

Thank you for considering our manuscript again. 

Response to editor’s comment 

Author’s response: We revised our manuscript based on journals style.

2. During our internal evaluation of the manuscript, we found significant text overlap between your submission and the following previously published works, some of which you are an author.

Author’s response: Thank you. We revise the manuscript in advance and re write the overlapped texts.

3. Thank you for stating the following financial disclosure as “None”. Please include your amended statements within your cover letter; we will change the online submission form on your behalf. 

Author’s response: Thank you. For this particular study the authors received no specific funding from any organization and we included this statement in the cover letter. 

4. Your ethics statement must appear in the Methods section of your manuscript. If your ethics statement is written in any section besides the Methods, please move it to the Methods section and delete it from any other section. Please also ensure that your ethics statement is included in your manuscript, as the ethics section of your online submission will not be published alongside your manuscript.

Author’s response: Thank you for the comment. We consider your comment and we put the ethics statement in the method section of the revised manuscript. Also, the ethics statement putted in the manuscript and in the online submission system is line. 

Response to reviewer’s comments 

Response to reviewer #1 

Dear reviewer we really thank you for your constructive comments and suggestion and we addressed the point you raised.

1. The manuscript need English editing.

 Author’s response: Thank you. We extensively edit our manuscript and grammatical errors are corrected. 

2. In table 3; it will be better if author add extra column for p value.

Author’s response: we accepted the comment and we have included the p- value in table 3 of the revised manuscript.

Response to reviewer #2 

1. I am doubting if it is eligible for the international readers of PLOSONE this as outcomes might be region specific.

Author’s response: Thank you for raising this important issue. This study is based on the nationally representative data with larger sample size from the nine regions and two-city administration of Ethiopia. The finding of this study might be very important for national policy makers to give appropriate intervention for this devastating public health problem which is especially common in young women. Also, the findings of this study might be important for other countries particularly sub-Saharan African countries and other lower and middle income countries which had almost similar socioeconomic and sociocultural patterns with our country. Moreover, other researchers and scholars might be used it as a base line for future studies. 

2. I do not understand why this study is needed, with regards to this age group, are young woman not part of the reproductive age group? What makes them different?

Author’s response: Dear reviewer thank you for your important concern. As you know, young women are parts of reproductive age women. However, young women and adolescent females are susceptible to anemia because of their biological demands for micronutrients associated with rapid physical growth and the depletion of these nutrients due to parasitic infestations. Anemia is also common in puberty due to the onset of menstruation, which alter an individual's iron status by generating more demand for iron, and blood loss. Moreover, anemia in young women is a serious condition, which impedes them from reaching their full potential by reducing educational achievement, labor productivity, as well as their cognitive capacity and affect their mental health, which might indirectly affect the future generation, because this women are responsible for the continuity of the generation. Studies also revealed that the prevalence (high prevalence) and the potential factors affecting anemia among reproductive age and young women might not be similar. Therefore, we believe that interventions on these age groups can have a great advantage to increase the academic achievements of women, their labor productivity, and their competitions in every aspect in the society. In addition, interventions at these age groups can prevent future occurrence of anemia and make them healthy throughout their reproductive age. Therefore, we aimed to assess anemia in in these age group women (15-24 years). 

3. Line 28: Is this for all of Africa or only sub-sharan Africa? As I believe the needs are not as much possible in northern Africa. Please rephrase.

Author’s response: Thank you. It is for most part of African country and modified in the revised manuscript.

4. Line 30: I do not understand the difference between young woman and woman in the reproductive age group. Is this not the same group? Why is there a difference?

Author’s response: Thank you for the important issue you raised. As we stated above, young women are parts of reproductive age group women. Nevertheless, they are at greater risks of anemia due to their extra needs of micro and macronutrients because of their rapid physiological growth at this age. Besides, if these young women are affected by this public health problem they might be loss their academic achievement and face mental and developmental problems, so they may lose their potential for contributing the development of the country as well as for their personal development. Because of this and the above-mentioned reasons, studying anemia in young population is very crucial.

5. Line 34 what kind of determinants?

Author’s response: Determinant indicate the individual and community level factors that affect anemia in young women and the statement indicating this is found everywhere in the revised version of our manuscript.

6. Line 35 MOR/ ICC; please no abbreviations in the abstract without explanation.

Author’s response: Thank you. Abbreviations are explained when first introduced in the abstract section.

7. Line 36 double (..) .

Author’s response: Thank you. We accepted the comment and we removed the double punctuation.

8. Line 36-38 this can be shortened not all needed in my perspective in the abstract. What is the definition of young woman’s? I can’t find that In the method.

Author’s response: young women that we used in this study is to mean women in the age group 15-24 years and we incorporated it in the method section of the Main manuscript (see in the method section line 132 in the revised manuscript). 

9. The terms young woman and adolescents are used all over the manuscript. However it is unclear what the exact definitions are.

Author’s response: Thank you. In general for our study young women were used for the analysis of this data. Usually adolescent women and young women have some difference. Adolescent women usually to mean women of age less than 20, might be from “10 to 19” or “13 to 19” years of old depending on different literatures. However, young women was to mean women from 15 to 24 years of age. Anemia may have a great health impact both in adolescent and young women, but the reason for using young women data for this study was since the EDHS data considers women in the age group 15- 49 years and studies , especially those done using DHS considers women in the age groups 15 to 24 as young women. The information for those adolescent women of less than 15 years are not available in the EDHS data and we prefer to use these age groups (15-24) as young women in this study.

10. Line 75-78 unclear sentences, which cut-off? Please define more specific.

Author’s response: Thank you. It is the WHO cut of point to define anemia as a public health problem. According to WHO, if the prevalence of anemia is greater than 5% it is considered as public health problem and we rephrase the statement in the revised version of our manuscript.

11. Line 79: Which population? Please define more specific

Author’s response: Thank you. This amount referred is from the total population of the world and we revised it in the manuscript.

12. Overall line 75-87 this alinea needs to be restructured, the data presenting on adolescents and young woman reproductive are not used in a structured way. Please rephrase and structure.

 Author’s response: Thank you for this very important concern. We take the comment and rephrase the full paragraph in the revised version of our manuscript.

13. Line 89-92: This is a line-up most likely from a previous study. Please summarize the only needed information.

Author’s response: Thank you. We accept the comment and we summarize the paragraph in the revised version of our manuscript.

14. Line 93: This is a repetition of what is said in the first alinea. Please rephrase and restructure.

Author’s response: Thank you. We amended the statement and removed repetitive sentence.

15. Line 95-96: The reason for his study is that most study look to the reproductive woman, it is unclear for me why this young woman are not part of that see previous comments. Can the author explain why this study is needed next to a study on woman in the reproductive age?

Author’s response: Thank you again. Studying prevalence and determinant of anemia in this young women had many clinical implications. The reason for studying anemia prevalence and associated factor in this age group is that the health impact of anemia differ among young women (15-24 years) and reproductive age (15-49 years) women due to the rapid growth and physiological changes occurring in young women resulting extra needs of macro and micronutrients. These extra needs of nutrients commonly result in nutritional deficiencies which are the common causes for anemia. Also the potential factors affecting anemia among reproductive age and young women are not similar as reported by different studies. For these reason and reasons what have indicated in the above comment we have conducted this study at these population subgroups (15-24 years). Also, unlike other study we have considered both the individual and community level factors.

16. Line 99: Abbreviation not explained.

Author’s response: Thank you. We expanded the abbreviation in the revised manuscript

Method

17. Line 107-111 can this be displayed in a graph that makes it easier for the reader to understand.

Author’s response: Thank you for your important issue and we display this information with graph to make it easier for reader in the revised manuscript (see Figure 1).

18. Line 113: add abbreviation (EDHS) as it used in the next sentence.

 Author’s response: Thank you. We added an abbreviation in the revised version of our manuscript.

19. Line 113-126 can this be added in a graph/ figure with a flow diagram. That makes it easier for the reader to oversee what happened.

Author’s response: Thank you for your important issue and we display this information with flow diagram to make it easier for reader (see figure 2). 

20. Outcome variable line 128: please define young woman.

Author’s response: Thank you. We define young women as those aged 15 to 24 years of age and included in the method section of the revised manuscript.

21. Line 129-131 I do not understand this sentence and I am questioning if the sentence is needed. Moreover there is no difference between in mild-moderate-and severe anemia in the study. So the author should remove this sentence.

 Author’s response: Thank you. From the EDHS report anemia is coded as no anemia, mild anemia, moderate and severe anemia, so we recode it as anemic and non-anemic. Also we edited and removed unnecessary information in the revised version of our manuscript.

22. Line 132: Reference 14; is this correct? And please add in writing which adjustment is done.

Author’s response: Thank you. The hemoglobin level was measured and adjusted for altitude using the adjustment formula (adjust = − 0.032*alt + 0.022*alt2 and adjHb = Hb - adjust (for adjust > 0). We used this, which was already provided in the EDHS data. We remove this reference here since it was not appropriately cited here.

23. Line 133: Reference 13; correct? This is the WHO definition of anemia please add the WHO as reference.

Author’s response: Thank you. We corrected the reference in the revised version of our manuscript. 

24. Independent variable: line 136-140: please add definitions of the variable or add a reference. For example what is wealth status? What includes media exposure?? What includes modern contraceptives? Please rephrase.

 Author’s response: Thank you for your important information and we put all the information about our independent variables in table (see table 1 on page18-20). 

25. Line 138 typo double (,,).

Author’s response: Thank you. We correct the double punctuation in the revised manuscript.

26. 148-149 Descriptive (no needed information) please take out.

Author’s response: Thank you. The statement about descriptive statics in the data management and analysis part is removed in the revised manuscript.

Results

27. Overall percentage can be given in one decimal.

Author’s response: Thank you. We put the overall prevalence with one decimal in the revised version of our manuscript.

28. Weighted sample not needed- please take out. The age range 15-24 can be taken out here, as this will be added to the method section

Author’s response: Thank you. We removed the phrase “weight sample” and the age range of 15-29 also taken out from the result section and we included it in the method section of our revised manuscript.

30. The structure of line 165-173: could the author structural outcomes better. As a suggestion group in individual medical- social-economics and community. And use this structure as well in table 1.

 Author’s response: Thank you. We restructure the result section as individual level and community level factors in the revised version of our manuscript. This correction also made to table 1.

31. Line 181-184: is this difference significant? Figure 1: what is the yellow line, please adjust figure

Author’s response: Thank you. The difference in the prevalence of anemia among young women higher in Somalia (56.80%) and Afar (43.93%) and lowest in Addis Ababa and this difference is found to be statically significant with non-overlapping 95%CI for the prevalence of anemia across this regions of Ethiopia. The figure is readjusted and here we incorporate the graph to indicate the variation in the range of prevalence of anemia among young women in different region of Ethiopia.

32. Line 196-213/ Table 3: regions: why is all compared to Addis? Nutrition status: why all compared to malnutrition and not to none? Why with occupation display no below yes. Same for contraceptive and media exposure. Please display in a constituent manner. Distance to health facility: not a big problem? What does that mean? How is this defined?? 

Author’s response: Thank you. Regarding the question about nutritional status we take the comment and amend it. Previously we were used malnutrion (underweight) as a reference, but we reanalyzed and women with normal BMI used as a reference to compare prevalence of anemia between people with normal nutritional status to those of malnutrition.

The variable occupation, contraceptive and media exposure no and yes displayed in the constituent manner in the revised version of our manuscript (see table 4 on page 23-25). 

 Regarding region, we compared anemia among young women across different region of Ethiopia and Addis Ababa was used as the reference, the reason for using Addis Ababa is that as compared to other regions of Ethiopia the prevalence of anemia was lowest in this city administration (Addis Ababa) compared to other regions. Therefore, we prefer the comparison of anemia in other regions to Addis Ababa. 

Concerning distant to health facility it is based on respondents response for the question whether the women perceive distance from the health facility as a big problem or not big problem (see table 1 on page 18-20).

Discussion 

33. I miss out in the discussion very clear statements why this outcome is 1) different than other woman previous investigated in Ethiopia? 2) is that surprising or not? What did others find in SSA for these age groups? Are you in line with that and if yes or no what are the difference? What are the clear limitations: limitations are stated very minimal. 

Author’s response: Thank you. We consider your comment in the revised manuscript (see the discussion section line 217-230 on page 10 and the strength and limitation section). 

34. Line 217: Armenia-Timor: why a comparison to this places?

Author’s response: Thank you. We compare our finding with study done in Armenia-Timor since it is part of low and middle income countries with similar finding to our finding. The socio economic status of this region might not be far from those of Ethiopia and comparison to this region may not have significant problem.

35. Line 218: our country, what was the rate?

 Author’s response: Thank you. We included the rate in the previous study conducted in our country and rewrite the statement in the revised version of our manuscript.

36. Line 219: Rate of comorbidity? This comment is based on which knowledge please explain

Author’s response: Thank you. Here the “rate of comorbidity” was to mean the increased rate of chronic diseases and other pathological condition which are the possible cause of anemia of chronic disease. 

37. Line 236-249: this alinea has a lot of repetitive statements. Please rephrase and structure

Author’s response: Thank you. We avoid the repetitive idea, statement and rephrase the paragraph in the revised version of our manuscript.

38. Line 247-249: Please add this to the section on rural areas in line 237-238

Author’s response: Thank you. We take the comment and included this statement as the explanation for the difference in anemia prevalence between urban and rural resident. Also it is an important justification for regional variation of anemia in Ethiopia.

39. Line 254

Author’s response: Thank you. We consider the comment and amended it in the revised version of our manuscript.

40. There is no data on age of first child, time between sampling and delivery. So major presumptions are made, but ideally this information would be available to say more about why are these women at risk. Our data on nutrition habits to underline the statements and suggestion. Moreover there is no data on malaria prevalence in this group. Or the availability of prophylaxis during pregnancy. All factors, which can be of major influence. The limitations need rephrasing and suggested limitations should be added.

Author’s response: thank you for raising this very important issue. For the analysis of this study we have used young women from 15 to 24 years of age. We have included women in this age group that might be pregnant or not and also this women might or might not have bear child so the variables like age at first child bearing and any prophylaxis given during pregnancy are not complete information for this analysis, which means all women included for this study not have given child birth or may not have history of pregnancy, so it is impossible to assess the above mentioned variables for this age group. Also data on malaria and nutritional habit are not available in the EDHS data. Finally, we have incorporated all these as limitation in the revised version of our manuscript.

Response to reviwer#3

1. This article covers an important global health concern. However, I believe the manuscript will benefit from a comprehensive language editing

Author’s response: we really thank you for your constructive comments. We extensively edited the spelling and grammatical errors.

2. Please pay particular attention to the following and review the sentences: Lines 25, 75, 95, 113, 129, 132,140, 150, 167, 170, 199, 229, 234, 238, 244, 253 and 257

Author’s response: Thank you for this important comment. We have looked and corrected all the spelling and grammatical errors line by line in the revised version of our manuscript. 

Response to reviewer #4

1. The discussion needs to be written with clarity and better understanding. Most of the explanations given for the reported findings are difficult to comprehend.

Author’s response: Thank you. We take look at the comment and we amend the discussion, particularly the explanation given for our finding (we extensively rephrased in the revised version of the manuscript). 

2. There are also too many grammatical and spelling errors. The manuscript will benefit from review by a native English speaker.

Author’s response: Thank you. We extensively edited the spelling and grammatical errors. Also the manuscript is checked by language experts in our institution. 

Response to reviewer #5

1. The first issue is that the paper would need substantial English language editing; there are many editorial errors in the paper.

Author’s response: Thank you. The manuscript is extensively reviewed by all the author for spelling and grammatical errors. Also language experts in our institution look and edited our manuscript.

Abstract

2. The statement that there is a high prevalence of anemia in young persons (line 29) as a justification for the study is contradictory because If there is already high prevalence of anemia among young women why have the authors conducted another study among this same population? 

Author’s response: Thank you. These studies were not based on a nationally representative data and most did not consider the community level factors. In addition, we clarified and rewrite the statement about the burden and magnitude of anemia among young women in the revised manuscript. 

3. The statement ‘The overall prevalence of anemia among young women were 21.7% (line 40) should read ‘…was 21.7%’

Author’s response: Thank you for your important issue. We removed the verb “were” which are grammatically incorrect and replaced by the correct verb “was”. 

Main manuscript

Background 

4. There is need to clarify that the vulnerable population being referred to in line 69 are female adolescents and young women; this point should be clarified throughout the manuscript.

Author’s response: Thank you for your important issue. We corrected all the venerable population is in this manuscript is about female adolescent and young women in the revised version of our manuscript. We made the correction throughout the manuscript.

5. The statement ‘over half of young women worldwide are suffered...’ line 75 should read ‘… have suffered’

Author’s response: Thank you. We replace the inappropriately used verb “are suffered” with grammatically correct verb “have suffered”.

 6. The statement ‘…and African as well...’ line 76 is not clear and should be revised.

Author’s response: Thank you. We take the comment and rephrased the indicated statement.

7. If the statement ‘approximately one quarter of adolescents…’ line 78 is referring to female adolescents this should be clarified

Author’s response: Thank you. We clarified this statement as it is being about female adolescent

8. There are many repetitive statements about the fact that sub-Sahara Africa has contributed to the high burden of anemia, this should be revised

Author’s response: Thank you for this important comment. We amended the repetitive statement that provides information about anemia burden and prevalence in sub Saharan Africa.

9. The authors have listed the socio-demographic factors (lines 89-92) that contribute to anemia in Ethiopia, but examples are not provided to illustrate this point. Some examples should be provided to illustrate this point.

Author’s response: Thank you. We have listed different factors including socio economic factor that may affect anemia among young women. Some of which was reported as a factor affecting anemia magnitude in previous Ethiopian studies and we have cited such studies in the revised manuscript. 

10. The statement ‘Due to rapid growth…’ line 93 is a repetition because this point has been made earlier

Author’s response: Thank you. We replace and rewrite the repeated word “due to rapid growth” in the revised version of our manuscript.

11. The sentence ‘…and up to our knowledge’ line 96 should read ‘…to our knowledge’

Author’s response: Thank you. The phrase ‘up to our knowledge’ replaced by ‘to our knowledge’ in the revised version of the manuscript. 

12. The data for the study were extracted from the 2016 EDHS; is this the latest survey in the country? Have there been other surveys since the one in 2016? If there has been a more recent survey, the authors need to justify why they have used the 2016 survey as source of data for this study. This is should be clarified.

Author’s response: Thank you. The data extracted for this survey was obtained from EDHS 2016 since this is the most recent survey conducted in our country.

Methods

13. An important information missing about the study area is the population of Ethiopia and more importantly the population of young persons in the country.

Author’s response: Thank you. We incorporate the missed information about study area. Also we have incorporated important information about recent population of Ethiopia in the revised version of our manuscript.

14. The reference to altitude and smoking (line 132) is not clear; how do these variables relate to the issue investigated.

Author’s response: Thank you. We take the comment and rephrased and justified the statement in the revised version of our manuscript. Also, the hemoglobin value recorded in the EDHS data was adjusted for altitude and smoking, since it is affected by the level of oxygen saturation at different altitude. Also Smoking increase hemoglobin (Hb) concentration mediated by exposure of carbon monoxide. Carbon monoxide binds to Hb to form carboxyhemoglobin, an inactive form of hemoglobin having no oxygen carrying capacity. Carboxyhemoglobin also shift the Hb dissociation curve in the left side, resulting in a reduction in ability of Hb to deliver oxygen to the tissue. To compensate the decreased oxygen delivering capacity, smokers maintain a higher hemoglobin level than non-smokers. So the minimum hemoglobin cutoff values should be adjusted for smokers to compensate for the masking effect of smoking on the detection of anemia and in the EDHS data the reported anemia level is already adjusted for this two variables. The reference cited here is in appropriately placed and we removed it. 

Results 

15. The statement ‘of women were not had work’ line 170-171 is not clear and should be revised

 Author’s response: Thank you. The statement about young women who had no work to mean that young women who were not have a job during the survey and we clarified in the revised version of our manuscript.

16. The word ‘afar’ line 183 should read ‘Afar’

Author’s response: Thank you. We amend “afar” with the correct spelling “Afar”.

 17. The use of the phrase ‘followers of Muslim and protestant religion’ line 203 are not clear. A better phrase may be people who practice Islam or Moslems and protestant Christians. This should be revised.

Author’s response: thank you. We rewrite the phrase “follower of Muslim and protestant religion” as people who practice Muslim and protestant Christian religion in the revised version of our manuscript.

18. The statement ‘… with their counterparts’ line 207 should read ‘with their counterparts who do not’.

Author’s response: Thank you. We address the issue in the revised version of our manuscript.

19. The figure showing prevalence of anemia should have a number and reference should be made to it in the text.

Author’s response: Thank you. We had made correction to the figure and cited in the main document of the manuscript.

Discussion 

20. I suggest that the authors provide the figure being referred to in the statement ‘but greater than the previous reported in our country’ line 218

 Author’s response: Thank you. We included the figure (number) which was reported in the previous study in our country in the revised version of our manuscript.

21. The authors should give examples of the food types being referred to in line 228 and support this statement with appropriate reference.

Author’s response: Thank you. We have incorporated some food type which might be restricted by some religion and culture that associated with malnutrition which is one possible cause of anemia. For example restriction of some red meat like; pork meat, fish meat, goat meat, bacon which are the possible source of iron by some culture and religion like by Islamic religion may be considered as a predisposing factor for anemia. We have included this culturally or religiously restricted food in the revised version of our manuscript.

 22. The statement i.e. barefooted line 241 should be revised to read walking barefooted

Author’s response: thank you. The word ‘barefooted’ revised to read as ‘walking barefooted’ in the revised version of the manuscript.

---

## [Decision Letter · Decision Letter 1]

29 Sep 2020

PONE-D-20-18515R1

Prevalence and determinants of Anemia among young (15-24 years) women in Ethiopia; A multilevel analysis of the 2016 Ethiopian Demographic and Health Survey data

PLOS ONE

Dear Dr. Worku,

Thank you for submitting your manuscript to PLOS ONE. After careful consideration, we feel that it has merit but does not fully meet PLOS ONE’s publication criteria as it currently stands. Therefore, we invite you to submit a revised version of the manuscript that addresses the points raised during the review process.

There are remaining comments from some reviewers that require your attention and must be addressed in your revised manuscript.

We look forward to receiving your revised manuscript.

Kind regards,

Frank T. Spradley

Academic Editor

PLOS ONE

Reviewers' comments:

Reviewer's Responses to Questions

**Comments to the Author**

1. If the authors have adequately addressed your comments raised in a previous round of review and you feel that this manuscript is now acceptable for publication, you may indicate that here to bypass the “Comments to the Author” section, enter your conflict of interest statement in the “Confidential to Editor” section, and submit your "Accept" recommendation.

Reviewer #1: All comments have been addressed

Reviewer #2: (No Response)

Reviewer #3: All comments have been addressed

Reviewer #4: All comments have been addressed

Reviewer #5: All comments have been addressed

2. Is the manuscript technically sound, and do the data support the conclusions?

Reviewer #1: Yes

Reviewer #2: Yes

Reviewer #3: Yes

Reviewer #4: Yes

Reviewer #5: Yes

3. Has the statistical analysis been performed appropriately and rigorously? 

Reviewer #1: Yes

Reviewer #2: I Don't Know

Reviewer #3: Yes

Reviewer #4: Yes

Reviewer #5: Yes

4. Have the authors made all data underlying the findings in their manuscript fully available?

Reviewer #1: Yes

Reviewer #2: Yes

Reviewer #3: Yes

Reviewer #4: Yes

Reviewer #5: Yes

5. Is the manuscript presented in an intelligible fashion and written in standard English?

Reviewer #1: Yes

Reviewer #2: Yes

Reviewer #3: Yes

Reviewer #4: Yes

Reviewer #5: Yes

6. Review Comments to the Author

Reviewer #1: This is a fine paper and will be useful to the field, and the results are interesting.

I don’t have any comment

Reviewer #2: Dear author,

Thanks for the major revisions to the manuscript. However I still have major concerns see them in detailed explained hereafter. But most important; I still mis out in your introduction and discussion. Why your study was essential to do? And what does it bring clearly new to this research field? Your methods and numbers are good but without that message very clear I am doubting for the eligibility for international readers. Please adjust that as you stated nicely in the explanations given to my comments.

• Reviewer 2 Comment 1 eligibility for international readers;

o Thanks for your clear response please add this information to the introduction

o Suggestion 1: change in the abstract line 27 particular in including

o Suggestion 2: line 99 add the age range of productive age (define)

o Suggestion 3: so explain why these two groups are different. Reproductive age and young woman and why your study is needed.

o Suggestion 4: line 100” determinates in Ethiopia” change to in Africa (or LMIC) including Ethiopia

• Reviewer 2 Comment 9: thanks for clarifying these definitions

o Suggestions: line 73 young woman and adolescents female. Change in including adolescent females

• Reviewer 2 Comment 15: see like the statement at comment 1. Clear clarification but please add this information to your introduction. To me it is still not clear why this study needed to be done. Please clarify clearly in the introduction and aim of the study.

• Reviewer 2 Comment 22: please state this calculation in the method section. Because you still used it and now it is taken out fully. That is incorrect. Please change.

• Reviewer 2 Comment 31:

o line 181-184: please add the CI to the groups you compare with as well and the p value. Now the sentence is not complete. Or take out all the CI and just give the P value. But the statement reads incorrect in the way it is stated now.

o Please add the CI to figure 3.

• Reviewer 2 Comment 32 : please add in section 181-184 to which area it was compared. Compared to addis… etc.

• Reviewer 2 Comment 34: This comparison to Timor is still out of place in my perspective. Please refrase. Either put this statement together with the comparison at line 225 or take out.

• Reviewer 2 Comment 35: thanks for this change. However to me it is still not clear why your study was important when you have these numbers already. Please rephrase and explain what your study showed more. Because in the way it reads now like you redid the study.

• Reviewer 2 Comment 40: line 275: add these variables which you stated in your explanation there aswell.

Reviewer #3: (No Response)

Reviewer #4: (No Response)

Reviewer #5: Prevalence and determinants of Anemia among young (15-24 years) women in Ethiopia; A multilevel analysis of the 2016 Ethiopian Demographic and Health Survey data

There is substantial improvement in the presentation of the revised manuscript. The authors have addressed all the issues and queries I raised in my previous review of the manuscript. However, in re-reading the manuscript, I detected some minor editorial errors which require the authors attention.

1. In line 80, the word ‘where’ should be replaced by the word ‘with’ to make the sentence clearer.

2. The statement on line 133 and 134 should be presented in past tense ‘for pregnant women…was considered as anemic…’

3. In line 165, the first letter of the university should start with a capital letter, to read ‘University of Gondar’; the authors should also clarify if this institution is in Ethiopia.

4. The statement on 226 needs revision; it should read ‘…greater than the 15.6% anemia found among young women in Rwanda’

5. The author should clarify which country is being referred to on line 227; is this Ethiopia or Rwanda? This should be revised.

6. The statement on line 262 should read ‘…which are common in lowland areas’

7. The word on line 330 should read ‘References’ not ‘reference’

7. PLOS authors have the option to publish the peer review history of their article (what does this mean?). If published, this will include your full peer review and any attached files.

Reviewer #1: **Yes: **Osama Mohammed Al-Amer

Reviewer #2: No

Reviewer #3: No

Reviewer #4: **Yes: **KEHINDE OKUNADE

Reviewer #5: **Yes: **Professor Ademola J. Ajuwon

---

## [Author Response · Author response to Decision Letter 1]

9 Oct 2020

We have included information about the importance of this study in the introduction section of the revised version of the manuscript as indicated by reviewers. Also we address all the reviewers comment accordingly.

---

## [Editor Report · Decision Letter 2]

14 Oct 2020

Prevalence and determinants of Anemia among young (15-24 years) women in Ethiopia; A multilevel analysis of the 2016 Ethiopian Demographic and Health Survey data

PONE-D-20-18515R2

Dear Dr. Worku,

We’re pleased to inform you that your manuscript has been judged scientifically suitable for publication and will be formally accepted for publication once it meets all outstanding technical requirements.

Kind regards,

Frank T. Spradley

Academic Editor

PLOS ONE

---

## [Editor Report · Acceptance letter]

19 Oct 2020

PONE-D-20-18515R2 

Prevalence and determinants of Anemia among young (15-24 years) women in Ethiopia: A multilevel analysis of the 2016 Ethiopian Demographic and Health Survey data 

Dear Dr. Worku:

I'm pleased to inform you that your manuscript has been deemed suitable for publication in PLOS ONE. Congratulations! Your manuscript is now with our production department. 

Kind regards, 

on behalf of

Dr. Frank T. Spradley 

Academic Editor

PLOS ONE